# Identification of Early Biomarkers of Mortality in COVID-19 Hospitalized Patients: A LASSO-Based Cox and Logistic Approach

**DOI:** 10.3390/v17030359

**Published:** 2025-02-28

**Authors:** Anna Maria Fratta Pasini, Chiara Stranieri, Edoardo Giuseppe Di Leo, Lorenzo Bertolone, Antonino Aparo, Fabiana Busti, Annalisa Castagna, Alice Vianello, Fabio Chesini, Simonetta Friso, Domenico Girelli, Luciano Cominacini

**Affiliations:** 1Department of Medicine, Section of Internal Medicine D, University of Verona, Policlinico G.B. Rossi, Piazzale L.A. Scuro 10, 37134 Verona, Italy; chiara.stranieri@univr.it (C.S.); edoardogiuseppe.dileo@univr.it (E.G.D.L.); lorenzo.bertolone2@gmail.com (L.B.); fabiana.busti@univr.it (F.B.); alice.vianello@univr.it (A.V.); fabio.chesini@univr.it (F.C.); domenico.girelli@univr.it (D.G.); luciano.cominacini@univr.it (L.C.); 2Interdepartmental Laboratory of Medical Research, Research Center LURM, University of Verona, 37134 Verona, Italy; antonino.aparo@univr.it; 3Department of Medicine, Section of Internal Medicine B, University of Verona, Policlinico G.B. Rossi, Piazzale L.A. Scuro 10, 37134 Verona, Italysimonetta.friso@univr.it (S.F.)

**Keywords:** COVID-19, early biomarkers, interleukin-10, lactate dehydrogenase

## Abstract

This study aimed to identify possible early biomarkers of mortality among clinical and biochemical parameters, iron metabolism parameters, and cytokines detected within 24 h from admission in hospitalized COVID-19 patients. We enrolled 80 hospitalized patients (40 survivors and 40 non-survivors) with COVID-19 pneumonia and acute respiratory failure. The median time from the onset of COVID-19 symptoms to hospital admission was lower in non-survivors than survivors (*p* < 0.05). Respiratory failure, expressed as the ratio of arterial oxygen partial pressure to the fraction of inspired oxygen (P/F), was more severe in non-survivors than survivors (*p* < 0.0001). Comorbidities were similar in both groups. Among biochemical parameters and cytokines, eGFR and interleukin (IL)-1β were found to be significantly lower (*p* < 0.05), while LDH, IL-10, and IL-8 were significantly higher in non-survivors than in survivors (*p* < 0.0005, *p* < 0.05 and *p* < 0.005, respectively). Among other parameters, LDH values distribution showed the most significant difference between study groups (*p* < 0.0001). LASSO feature selection combined with Cox proportional hazards and logistic regression models was applied to identify features distinguishing between survivors and non-survivors. Both approaches highlighted LDH as the strongest predictor, with IL-22 and creatinine emerging in the Cox model, while IL-10, eGFR, and creatinine were influential in the logistic model (AUC = 0.744 for Cox, 0.723 for logistic regression). In a similar manner, we applied linear regression for predicting LDH levels, identifying the P/F ratio as the top predictor, followed by IL-10 and eGFR (NRMSE = 0.128). Collectively, these findings underscore LDH’s critical role in mortality prediction, with P/F and IL-10 as key determinants of LDH increases in this Italian COVID-19 cohort.

## 1. Introduction

The pandemic coronavirus disease 2019 (COVID-19) caused by the novel coronavirus, severe acute respiratory syndrome Coronavirus-2 (SARS-CoV-2), has become one of the major global health crises of the 21st century [1]. Patients infected with SARS-CoV-2 suffer from different symptoms and forms of the disease, which may depend on both patient-related factors, infection, and the virus itself. Although many patients are asymptomatic or have mild symptoms such as fever, fatigue, and dry cough, a few cases advance to a more severe form of the illness, with a high risk of death associated with respiratory failure, circulatory failure, and multiple organ failure [2]. Even if a large number of studies have been published so far, it is still unclear if comorbidities and certain risk factors increase the risk of death from COVID-19 and why some patients can oppose the SARS-CoV-2 infection effectively while others require hospitalization. Acute lung injury and acute respiratory distress syndrome (ARDS) appear in more severe COVID-19 diseases, which can induce morbidity and mortality due to pneumonia and inflammation caused by damage to the alveolar lumen [3,4]. Angiotensin-converting enzyme 2 (ACE2) offers an access receptor for SARS-CoV-2 in humans by binding to the viral membrane spike (S) protein [5,6]. ACE2 is ubiquitously expressed, with the highest levels in the epithelial cells of the lung, kidney, and cardiomyocytes [7,8]. Within the lung, the epithelial cells facing the lower airways are the primary viral targets [9,10] and undergo cell death as part of the viral replication cycle [11]. Recently, it has been reported that regulated necrosis is the predominant form of alveolar epithelial cell death in COVID-19-induced ARDS [12]. Regulated necrosis has come out as a new concept: it depicts different signaling pathways, such as kinase-mediated necroptosis, gasdermin-mediated pyroptosis downstream of inflammasomes, and an iron-catalyzed mechanism called ferroptosis [13,14,15,16,17]. In general, in viral infections and SARS-CoV-2 in particular, at least in the early stage, regulated necrosis is thought to inhibit virus invasion by killing cells [12]. Although regulated necrosis is involved in limiting the spread of SARS-CoV-2 infection, later in the disease, this mechanism has highly proinflammatory sequels [17]. In addition, amplifying inflammatory reactions leads to necroptosis and pyroptosis of macrophages, which worsen lymphopenia by killing lymphocytes, limiting the immune pertinent response [18]. Moreover, regulated necrosis discharges a considerable amount of pathogen-associated molecular patterns (PAMPs) and damage-associated molecular patterns (DAMPs), such as cytokines, SARS-CoV-2 particles, chemokines, lactate dehydrogenase (LDH), adenosine triphosphate, and reactive oxygen species. Eventually, this discharge further promotes an inflammatory cascade reaction, which ultimately causes cytokine storm in host cells and cytokine release syndrome in humans [19,20,21,22]. Therefore, in this study, we first aimed to identify possible early biomarkers of in-hospital mortality among clinical and biochemical parameters as well as iron metabolism parameters and cytokines detected within 24 h from the admission in hospitalized non-survivors and survivors of COVID-19 patients. Since our data showed that LDH was the most critical feature in discriminating between survivors and non-survivors, we further analyzed its correlations with the other parameters evaluated in this study. Then, to identify the most relevant clinical and biochemical features in predicting mortality and LDH levels in COVID-19 patients, we applied a combination of feature selection and predictive modeling techniques, integrating LASSO feature selection with Cox proportional hazards regression and logistic regression. For mortality prediction, both Cox proportional hazards regression and logistic regression were tested, with Cox regression yielding slightly superior results. LASSO was employed in both regression models for feature selection, helping to reduce the predictor set to the most significant features, which improved the model’s performance and interpretability. For predicting LDH levels, we combined LASSO with linear regression, allowing for effective feature selection and robust prediction of LDH outcomes based on the most relevant predictors. This approach provided a refined understanding of factors influencing LDH levels and mortality, offering valuable insights into disease mechanisms and patient prognosis.

## 2. Materials and Methods

### 2.1. Study Population and Design

The study was approved by the Ethical Committee of the Azienda Ospedaliera Universitaria Integrata Verona (prot. n. 3982CESC). Due to the exceptional pandemic circumstances, the Ethical Committee waived the requirement of written informed consent for participation in observational and non-interventional studies. Oral informed consent with annotation in the medical records was considered sufficient. It is an observational, retrospective study using plasma samples of hospitalized COVID-19 patients stored at −80 °C in an authorized biobank. The diagnosis of SARS-CoV-2 infection was confirmed on real-time reverse-transcriptase-polymerase-chain-reaction. Pneumonia was assessed by computed tomography. Patients with clinical features and imaging consistent with COVID-19 pneumonia hospitalized at the Verona University Hospital (Italy) from April 2020 to May 2021 were included in the study.

The inclusion criteria were:Presence of acute respiratory failure;Blood samples obtained within 24 h of hospital admission.

Exclusion criteria included the following: Hepatitis B virus and Human Immunodeficiency Virus infection, active cancer, immunosuppression with high-dose steroids, ongoing chemotherapy, and pregnancy. Forty patients discharged alive were matched for age and sex with 40 patients who died during hospitalization. Demographic characteristics, past medical history, biochemical parameters, and arterial blood gas tests were collected at hospital admission. Information about in-hospital evolution was retrospectively collected from the medical records.

### 2.2. Plasma Sample Collection and Biochemical Parameter Measurement

Plasma samples were collected after whole blood centrifugation at 400× *g* for 10 min. The undiluted plasma was then transferred to 10 mL polypropylene tubes (Corning, Tamaulipas, Mexico), aliquoted, and stored at −80 °C for subsequent analysis. Laboratory assessments comprised complete blood count, iron metabolism biomarkers, liver and renal function, C-reactive protein (CRP), and LDH.

The levels of the following circulating cytokines and chemokines were investigated by using Simple Plex assays run on the ELLA microfluidic immunoassay system (ProteinSimple, San Jose, CA, USA), according to the manufacturers’ instructions: interferon-gamma (IFNγ), interleukin-1 beta (IL-1β), interleukin-1 receptor antagonist (IL-1Ra), interleukin-22 (IL-22), interleukin-6 (IL-6), interleukin-8 (IL-8), interleukin 10 (IL-10), and tumor necrosis factor-alpha (TNF-α). Calibration of Ella was performed using the in-cartridge factory standard curve, and plasma samples were measured with a 1:3 dilution in Sample Diluent (ProteinSimple, San Jose, CA, USA). A single well was used for each sample as triplicate assays are automatically performed in the Simple Plex assay microfluidic platform. The lower limit of quantification was 0.28 pg/mL; the upper limit was 2652 pg/mL.

### 2.3. Statistical Analysis

Statistical analysis was performed using R version 4.3 (R Foundation for Statistical Computing, Vienna, Austria). Data preprocessing involved Z-score normalization to standardize patient characteristic distributions across the dataset. This normalization technique, crucial for mitigating the potential effects of differing measurement scales and ensuring comparability between variables, allows us to express each data point in terms of standard deviations from the mean. The normality of the data distribution was assessed using the Shapiro–Wilk test [23]. The test revealed that only a few parameters (age, hemoglobin, hematocrit, red blood cells, and LDH) followed a normal distribution. Given the non-normal distribution of the majority of the data, the Wilcoxon test [24] was employed to compare differences between survivor and non-survivor patient groups across demographic, clinical, and biochemical characteristics. A significance threshold of 0.05 was applied for determining statistical significance. Spearman correlation analysis was then conducted to explore relationships between demographic, clinical, and biochemical characteristics, aiming to elucidate potential associations among these variables and further enhance our understanding of their interplay in the context of COVID-19 patient outcomes. Kaplan–Meier (KM) survival curves were constructed for each clinical and biochemical feature to explore their univariate association with mortality. Patients were dichotomized in high vs. low groups based on the median value of the parameter under consideration. The log-rank test (Mantel–Cox) was used to assess differences in survival curves between the two groups. Statistical significance was set at *p* ≤ 0.05.

As reported in Appendix A, not all characteristics are known for some patients. To facilitate the application of artificial intelligence techniques utilizing all clinical features and subjects, applying imputation on missing data was necessary. The Miss Forest algorithm [25] was employed for this purpose. Miss Forest employs a machine learning algorithm that iteratively estimates missing values based on observed data patterns, starting with the features with the lowest proportion of missing values. Several imputation methods were evaluated, and further details on these methods are provided in the Appendix A. Based on the comparison, Miss Forest demonstrated superior performance and was chosen to impute missing data before proceeding to subsequent analyses. Following data imputation, feature selection analysis was conducted to identify a predictive signature—a set of characteristics associated with patient outcomes using LASSO [26] in combination with both Cox proportional hazards regression [27] and logistic regression. The primary goal of this multivariate approach was to identify features that were especially informative for two main purposes: (1) effectively distinguishing between survivors and non-survivors, and (2) predicting the value of LDH, a critical marker in COVID-19 patients. By applying LASSO, we aimed to reduce the predictor set to the most significant features, improving both predictive performance and interpretability.

Multiple LASSO configurations were tested to identify optimal predictors by adjusting the penalization parameter (lambda) during cross-validation, with the evaluation metrics provided in the Appendix A. For mortality prediction, both Cox proportional hazards regression and logistic regression were combined with LASSO, with model performance evaluated through AUC. For LDH prediction, LASSO was combined with linear regression, with performance assessed using Normalized Root Mean Squared Error (NRMSE). Once the best LASSO configuration was determined, 1000 iterations were performed to evaluate model stability. In each iteration, highly correlated features were removed, LASSO was applied, and the selected features were used for Cox, logistic, and linear regression models. Model performance metrics, AUC for mortality prediction, and NRMSE for LDH prediction were averaged across iterations using an 80/20 train-test split for consistent model evaluation. To quantify each feature’s predictive contribution, permutation importance was calculated by measuring changes in AUC or NRMSE when feature values were permuted in the test set. Only features with a consistent positive predictive impact across 1000 iterations were retained, ensuring the final models focused on the most robust and informative predictors. To evaluate the models’ performance and generalizability, mean ROC curves for mortality prediction and mean NRMSE values for LDH prediction were constructed based on the retained features from 1000 iterations. This approach confirmed the robustness of the selected features and the predictive capacity of the models.

To ensure the validity of the Cox proportional hazards model, we tested the proportional hazards assumption using the Schoenfeld residual test. This test assesses whether residuals from a Cox regression model are independent of time, as required for the proportional hazards assumption to hold. The test was applied to all variables included in the final Cox model, and global as well as individual test statistics were evaluated. A *p*-value > 0.05 indicates no significant violation of the proportional hazards’ assumption.

## 3. Results

Following the inclusion criteria, 80 hospitalized patients (40 survivors and 40 non-survivors) with COVID-19 pneumonia and acute respiratory failure were enrolled in this study. The Wilcoxon test was utilized to assess disparities between survivor and non-survivor patient cohorts across various demographic, clinical, and biochemical parameters. Table 1 summarizes the demographic, clinical, and biochemical data of the two groups of COVID-19 patients. Median time from the onset of COVID-19 symptoms to hospital admission was lower in non-survivors than in survivors, 5.5 days [2–7.7] vs. 8 days [5–13] (*p* < 0.05), respectively. Respiratory failure expressed as P/F was more severe in non-survivor than survivor patients with median P/F 224 [166.5–252] vs. 281 [229–314] (*p* < 0.0001), respectively. All the patients required supplementary O_2_ in different modalities during hospitalization: standard O_2_ supplementation (oxygen delivered through a nasal cannula, Venturi mask, or reservoir mask) or non-invasive ventilatory support or orotracheal intubation. Admission to the Intensive Care Unit (ICU) and the need for orotracheal intubation were more frequent in non-survivors than in survivor patients (*p* < 0.005). According to the inclusion criteria, the demographic characteristics were similar in the two groups. In addition, comorbidities, such as hypertension, diabetes mellitus, and chronic kidney disease, were similarly distributed in both groups, suggesting that comorbidities had no significant impact on the clinical outcome of our patients (Table 1). Biochemical parameters such as creatinine, AST, and ALT were similar and in normal range in both groups, whereas estimated glomerular filtration rate (eGFR) was significantly lower in non-survivors than in survivor patients (*p* < 0.05). Furthermore, LDH was significantly higher in non-survivors than in survivors (*p* < 0.0005). Except for the significant reduction in the number of lymphocytes and monocytes (*p*, respectively, <0.05), blood count parameters did not differ significantly between groups even if there was a trend towards lower hematocrit, hemoglobin (Hb), platelets, and higher red blood cell distribution width (RDW) levels in the non-survivor group, likely reflecting a tendency toward a more severe COVID-19 disease in non-survivors. As for the iron metabolism parameters, ferritin was similarly increased, whereas iron, transferrin, and transferrin saturation were similarly decreased in both groups. CRP levels were elevated in both groups.

Table 2 summarizes plasma levels of the tested cytokines IFNγ, IL-1β, IL-1Ra, IL-10, IL-22, IL-8 IL-6, and TNF-α in COVID-19 survivors and non-survivors and reports their statistical significance. IFNγ, IL-1Ra, IL-22, IL-6, and TNF-α did not differ significantly in the two groups of patients. IL-1β was lower in non-survivors than in survivors (*p* < 0.05). On the contrary, IL-10 and IL-8 were higher in non-survivors than in survivors (*p* < 0.05 and *p* < 0.005, respectively). Figure 1 shows the boxplots of clinical and biochemical characteristics deemed different between study groups through the Wilcoxon test. Among other parameters, LDH showed the most significant difference between survivors and non-survivors (*p* < 0.0001). The correlation matrix in Figure 2 indicates the significant correlations between the biochemical parameters of Table 1 and the cytokines of Table 2. Wilcoxon test results comparing demographic, clinical, and biochemical features between survivor and non-survivor patient groups are shown in Appendix A.

Given the relevant difference in LDH values distribution between study groups, we then focused on the possible correlations involving LDH (Table 3). One of the strongest correlations emerging from the correlation matrix is between LDH and P/F (r^2^= −0.792, *p* = 0.0 × 10^0^). LDH was also correlated with some cytokines and additional inflammatory circulating markers; a positive correlation was observed with IFNγ, IL-10, IL-1Ra, IL-6, CRP, and LDH (respectively, r = 0.361, *p* < 0.0001; r = 0.606, *p* < 0.00000001; r = 0.340, *p* < 0.001; r = 0.345, *p* < 0.001, r = 0.323, *p* < 0.006). Conversely, IL-1β was inversely correlated with LDH (r = −0.276, *p* < 0.01). Among the other biochemical markers, we observed an inverse correlation between the number of lymphocytes, iron, and LDH (r = −0.363, *p* < 0.001; r = −0.288, *p* < 0.001). On the contrary, a direct correlation was found between hepcidin and LDH (r = 0.33, *p* < 0.003). Table 3 shows the significant Spearman correlations involving LDH, while Appendix A shows the correlations between all the parameters considering the data of all patients.

Following the Wilcoxon and Spearman analyses, we evaluated the univariate association of each clinical and biochemical feature with patient survival over time using KM curves. For each variable, patients were stratified into high vs. low groups based on the median baseline value. Among all features considered, LDH showed the most consistent and significant differences in survival (log-rank *p* = 0.0025).

Figure 3 illustrates the KM curve for LDH, highlighting markedly reduced survival probabilities in patients with higher LDH levels. These findings confirm the important prognostic role of LDH at hospital admission. We then explored its combined impact, along with other variables, in our multivariate Cox analysis, as described in the following section.

Subsequently, we performed a machine learning-based analysis to identify key predictors of mortality and LDH levels in COVID-19 patients. This approach involved applying LASSO-based feature selection combined with different regression models to address distinct predictive objectives: LASSO with Cox regression and LASSO with logistic regression for mortality prediction (comparing their performance) and LASSO with linear regression for predicting LDH levels. We tested various LASSO configurations to identify the optimal setup, detailed in the Appendix A, ultimately selecting the deviance-optimized configuration (LASSO Deviance) as the most effective for both Cox and logistic regression due to its robust performance on the Reduced Dataset, from which highly correlated features were removed features (defined as those with Spearman correlation coefficient > 0.8). Specifically, for the LASSO–Cox regression model, after identifying the best models, a permutation importance analysis was conducted to quantify the contribution of each feature to the predictive accuracy of the final models. Before proceeding with the Cox proportional hazards analysis, we assessed the proportional hazards assumption using the Schoenfeld residual test. None of the selected predictors violated this assumption (*p* > 0.05 for all variables; see Appendix A). This confirms that the Cox model is appropriate for our dataset. As illustrated in Figure 4, the analysis identified LDH as the most influential predictor, with a mean AUC drop of approximately 0.10, underscoring its strong association with mortality in this cohort. Following LDH, IL-22 and creatinine had substantial impacts on model performance, with mean AUC drops of around 0.04 and 0.03, respectively, highlighting their relevance in predicting disease severity. Additional variables, including eGFR (mean AUC drop of 0.03), hepcidin (Hepc) (0.02), hemoglobin (Hb) (0.01), IL-10 (0.01), IFNγ (0.01), and monocytes (0.01), contributed as secondary predictors. Together, these features enhance the model’s robustness by providing complementary insights into disease progression. Features with minimal or negative mean AUC drops, such as platelets (−0.05), WBC (−0.04), and iron (Fe) (−0.02), were less influential, suggesting that they do not independently contribute to mortality prediction. These findings allowed the model to focus on the most critical predictors, improving interpretability and supporting a more streamlined approach that emphasizes the strongest independent predictors of mortality in COVID-19 patients.

The permutation importance analysis clarified the specific contribution of each feature to the predictive strength of the model, allowing us to refine the variable set to only the most impactful predictors. With these influential features identified, we pro-ceeded to evaluate the model’s overall performance and generalizability. To achieve this, we generated an average ROC curve based on 1000 iterations, using the final predictive model constructed with the most critical variables identified by the LASSO Deviance configuration on the Reduced Dataset. This averaged ROC curve, shown in Figure 5, provides a robust measure of the model’s ability to generalize to unseen data. The high mean AUC of 0.744 across test sets underscores the predictive strength and reliability of the selected features, indicating consistent performance in identifying mortality-associated factors with high sensitivity and specificity. Together, the insights from the permutation importance analysis and the consistent ROC performance establish a refined model with both interpretative clarity and robust predictive capacity, making it suitable for potential clinical applications in prognostic assessment. This final model, derived from features with demonstrated predictive relevance, offers a streamlined and interpretable approach to mortality prediction in clinical settings.

In addition to the LASSO–Cox analysis, we also applied LASSO combined with logistic regression for mortality prediction. While both approaches identified similar key features, logistic regression produced a slightly lower mean AUC of 0.723, compared to 0.744 for Cox regression. As shown in Figure 6, LDH remained the most influential predictor in the LASSO–logistic model, followed by IL-10, eGFR, and creatinine. These predictors also had substantial impacts in the Cox model, underscoring their consistent relevance across different modeling approaches. The ROC curve in Figure 7 illustrates the predictive performance of the LASSO–logistic model, showing robust sensitivity and specificity, though slightly less effective than the Cox-based model.

In predicting LDH levels, we employed LASSO for feature selection combined with linear regression to ensure model simplicity and avoid overfitting. The optimal model was determined through cross-validation, using a reduced dataset where highly correlated features were excluded (Spearman correlation coefficient > 0.8). This approach allowed us to focus on the most relevant predictors of LDH while eliminating redundancy. The predictive performance of the LASSO–linear regression model was evaluated using the Normalized Root Mean Square Error (NRMSE). Across 1000 iterations, the mean NRMSE for the training set was 0.066 (SD = 0.007), indicating good fit to the training data. However, when applied to the test set, the mean NRMSE was 0.128 (SD = 0.056), reflecting moderate predictive accuracy on unseen data. As shown in Figure 8, the LASSO–linear regression analysis identified a set of key predictors for LDH levels. The most influential predictor was the P/F ratio (PaO_2_/FiO_2_), which had a significantly higher importance score compared to other features, reinforcing its known clinical relevance in assessing respiratory function in COVID-19 patients. IL-10 was also a significant contributor to the prediction of LDH levels, indicating the strong association of this inflammatory marker with disease severity. Other notable predictors included eGFR, RDW, IL-1Ra, number of monocytes, hepcidin, and age, each contributing moderately to the predictive model.

The feature importance analysis underscored the strong relationship between inflammation markers (such as IL-10) and LDH levels, as well as the critical role of renal function markers (eGFR and creatinine). On the other hand, features like WBC, TNF-α, IL-22, and others showed lower importance, suggesting that they contributed less to the model’s ability to predict LDH levels. Overall, the use of LASSO with linear regression provided a clear and interpretable model for predicting LDH levels, with a subset of clinical and biochemical variables showing strong predictive importance. These findings contribute to a deeper understanding of the factors influencing LDH levels in COVID-19 patients, offering insights into the underlying mechanisms of disease severity.

## 4. Discussion

In this study, we first compared some clinical and biochemical parameters at hospital admission in non-survivor and survivor patients with COVID-19 pneumonia and acute respiratory failure. As for the clinical features, the onset of symptoms and P/F were the only significantly different features between the two groups, the latter being significantly lower in non-survivors. This was expected, P/F being a marker of respiratory failure severity. The most significant difference among biochemical parameters was found for LDH plasma levels, which were significantly higher in non-survivors compared to survivors. Our results agree with previous studies reporting increased LDH plasma concentrations in patients with COVID-19 [28,29,30,31,32,33]. It has been suggested that the rise in circulating LDH likely indicates an expansion of the activity and extent of organ injury and inflammatory cell death [28,29]. This notion is supported by the fact that LDH is a general indicator of tissue and cell damage, and that in COVID-19 there is extensive cell death among inflammatory cells, alveolar epithelial cells, and endothelial cells of the lungs and kidneys [34,35]. In this context, it has been reported that LDH is strongly correlated with COVID-19 pneumonia progression as documented by computed tomography [33]. The increase in LDH is likely related to cell death as part of the viral replication cycle [11]. In SARS-CoV-2 infection, several types of regulated cell death, such as necroptosis, pyroptosis, and ferroptosis, are thought to inhibit virus invasion by killing host cells [12,36]. In this way and in contrast to apoptosis, regulated cell death causes the release of DAMPs such as high-mobility group box-1 and others, causing, among others, the release of LDH from dead cells [37,38,39]. The response to regulated necrosis is connected to vascular leakage inside the alveoli, a phenomenon that causes local inflammation and enrolls immune cells from the blood into the lungs to remove viruses and disrupt virus-infected cells [19,40]. During the inflammatory phase, the disease can quickly move to acute respiratory distress syndrome (ARDS), a hyperinflammatory condition characterized by cytokine storm, associated with multiorgan dysfunction and a high mortality rate [21,41]. This study reported that the increase in LDH was inversely correlated with P/F as an indicator of respiratory failure. The results of this study also show a reduction in lymphocyte and monocyte numbers in COVID-19 patients, which was more significant in non-survivors than in survivors. Lymphopenia is a well-established hallmark of severe COVID-19, characterized by an absolute reduction in CD4+ and especially in CD8+ T cells [41,42,43]. Without a strong CD4+ T cell activation, B cells produce an antibody return that is insufficient to counteract SARS-CoV-2. Enhancement of exhausted T cells indicates a reduced proliferation and activity of CD8+ T cells [44,45]. Likewise, natural killer cells exhibit impaired cytotoxic activity, resulting in continuous viral shedding with further macrophage and neutrophil activation, which causes huge production of cytokines [46,47]. As for the reduction in circulating monocytes, it was reported that monocytes from patients with COVID-19 exhibited no increment in staining for annexin V, a marker of damaged plasma membranes, which is a pattern characteristic of pyroptosis or other forms of programmed necrosis [47]. This study’s data also indicate an overall increase in the cytokines and chemokines analyzed in both groups. However, only IL-1β, IL-10, and IL-8 resulted significantly different in non-survivors compared to survivors. IL-10 and IL-8 were higher, while IL-1β was lower in non-survivors than in survivors. IL-6 values were more elevated in non-survivors than in survivors, but the difference was barely below the statistical significance. As for plasma IL-1β levels in COVID-19 patients, only an initial report showed a marked increase in IL-1β in patients exposed to the Wuhan Huanan Seafood Market [41]. On the contrary, most of the studies did not find any difference in IL-1β between severe versus moderate cases or even did not report any statistically significant increase in serum IL-1β levels [47,48,49,50,51,52]. These data are perhaps not unexpected considering the remarkably short half-life of IL-1β [53,54]. Cytokine release and successive recruitment of immune cells can be repeatedly reiterated, giving rise to hyperinflammatory settings carrying a high cargo of cytokine-secreting cells associated with severe disease [55]. This impressive infiltration of immune cells and release of cytokines in lung tissue is a distinctive feature of ARDS and may be partly responsible for severe lung damage in patients with COVID-19 [55]. Giving that in our study some variables were found to be correlated with LDH, which in turns explains about 73% of respiratory failure, we can hypothesize that at least some of these parameters participate in the mechanisms leading to regulated cell death of lung cells. Thus, we applied LASSO with Cox regression and LASSO with logistic regression for mortality prediction to identify the most important demographic, clinical, and biochemical predictors of survival. By using both models, LDH emerges as the most discriminating feature compared to other variables. This result aligns with a series of papers summarized in a recent large meta-analysis [32]. LDH elevation in our COVID-19 patients likely indicates tissue injuries and, in particular, lung damage, as it emerges from the inverse correlation between LDH and P/F. Furthermore, when LASSO with Cox regression model was applied, IL-22 emerged as the second most important predictor of mortality while in the LASSO with logistic regression model, IL-10 turned up as the second most significant predictor of mortality. Even if on the basis of the present data we cannot fully explain this contradictory results, one possible explanation could derive from the fact that both cytokines belong to the IL-10 family and that they share the same receptor and activities. Like all members of this family in fact, IL-22 acts via a transmembrane receptor complex that consists of two different subunits: IL-22 receptor subunit 1 (IL-22R1) and IL-10R2. In particular, IL-22 mediates its cellular effects via a heterodimeric receptor complex composed of IL-22R1 and IL-10R2. The components of the heterodimeric IL-22R complex are also used by other cytokines of the IL-10 family. IL-10R2, for instance, mediates the effects of IL-10 (in a complex with IL-10R1) [56]. It must be underlined that IL-22 is unusual among most interleukins because it does not directly regulate the function of immune cells. Rather, IL-22 targets cells at outer-body barriers, such as the skin and tissues of the digestive and respiratory systems, as well as cells of the pancreas, liver, kidney, and joints [57,58,59,60]. Numerous studies have suggested that IL-22 plays a crucial role in anti-viral infections through significantly ameliorating the immune cell-mediated inflammatory responses, and reducing tissue injury as well as further promoting epithelial repair and regeneration [61,62,63] In this context, there are data suggesting that the involvement of the IL-22R1/IL-22 axis could be protective at the beginning of SARS-CoV-2 infection but could shift to a detrimental response over time and therefore predict disease severity [64,65].

As for IL-10, a distinctive characteristic of the cytokine storm in COVID-19 is the noticeable elevation of IL-10 [66]. The latter was believed to represent a negative feedback machinery used to arrest inflammation. However, several lines of evidence indicate that a remarkable early increase in IL-10 may have a pathological role in COVID-19 severity [66]. Our findings agree with a recent meta-analysis including non-severe and severe COVID-19 patients, suggesting that IL-10 precisely predicted disease severity and mortality [67].

In both LASSO models, eGFR and creatinine emerged as further subsidiary predictors of mortality. It is well known that, in addition to the respiratory tract and heart injuries, the kidneys are among the most common targets of SARS-CoV-2 [68]. Data from autoptic series indicate that SARS-CoV-2 is present in many kidney compartments, such as the renal parenchyma, glomerular epithelial, endothelial, and tubular cells [68]. From a clinical point of view, the most frequent kidney damage is acute kidney injury (AKI), whose pathogenesis is believed to be multifactorial. In fact, in addition to the direct effects of SARS-CoV-2 via ACE2, which is extensively present in different regions of the kidney [69], many indirect effects relate to secondary kidney damage [70]. As a matter of fact, renal dysfunction ranging from mild proteinuria and hematuria to overt AKI was significantly prevalent from the beginning of the SARS-CoV-2 pandemic [43,71]. Our results showing that eGFR can predict mortality in COVID-19 patients agree with the data of Cei et al. [72], who demonstrated that early reduction in estimated eGFR predicts poor outcomes in acutely ill hospitalized COVID-19 patients first admitted to regular medical wards. As for creatinine, an increase of as little as 0.3 mg/dL could signify a great reduction in GFR so that creatinine serum levels in the normal range or with minor variations could hide dangerous GFR alteration. In fact, from our results it emerged that a significant reduction in creatinine of 0.2 mg/mL corresponded to a significant decrease in eGFR from 81.5 to 60.0 mL/min/1.73 m^2^). Accordingly, measuring both eGFR and creatinine serum level changes is strongly indicated mostly for checking out AKI in patients with COVID-19 [73]. Additional variables in the LASSO with Cox regression model, including hepcidin, hemoglobin, IL-10, IFNγ, number of monocytes, IL-6, number of lymphocytes, and age contributed as negligible secondary predictors. Similarly, the secondary contribution of additional variables in the LASSO with logistic regression model including IFNγ, number of monocytes, and age was small.

Finally, the LASSO–linear regression analysis identified a set of key predictors for LDH levels. The most influential predictor was the P/F, which had a significantly higher importance score compared to other features, reinforcing its known clinical relevance in assessing respiratory function in COVID-19 patients. IL-10 was also a significant contributor to the prediction of LDH levels, indicating the strong association of this inflammatory marker with disease severity. Other notable predictors included eGFR, RDW, IL-1Ra, number of monocytes, hepcidin, and age, each contributing moderately to the predictive model. In our study, therefore, P/F and IL-10 emerged as strong predictors of LDH plasma concentrations. While the relationship between P/F and LDH is easy to understand since LDH is a marker of the mechanisms leading to regulated necrosis also of lung cells and, therefore, of respiratory failure, the connection with IL-10 is more complex to explain. Although the precise mechanisms contributing to lung injuries in COVID-19 are not fully understood, it is generally accepted that cytokine storm plays an important role [74,75]. The prolonged presence of cytokines and chemokines induces massive accumulation of neutrophils, eosinophils, and natural killer cells [75] that secrete a series of cytotoxic substances leading to alveolar injuries and respiratory failure [75]. IL-10 is a pleiotropic cytokine that primarily acts as an anti-inflammatory cytokine and preserves the body from an uncontrolled immune response [64,65,66,76,77]. However, it has been hypothesized that IL-10 might also play a pathological role in COVID-19 disease progression [66]. In this context, Lu et al. [66] suggested that early induction of IL-10 upon SARS-CoV-2 infection may act as an anti-inflammatory cytokine able to counteract the inflammation caused by other proinflammatory mediators. Afterward, as the production of IL-10 grows, it may work as an immune-activating/proinflammatory factor, triggering the production of other proinflammatory mediators. Furthermore, since IL-10 directly enlarges CD8+ T cells, hyperstimulation of adaptive immunity in COVID-19 patients may further worsen disease severity [78]. The combined effect of IL-10 in promoting systemic hyperinflammation and stimulating T cell activation may potentially damage multiple organs throughout the body and, therefore, explain why IL-10 is a strong predictor of LDH increase in our patients infected by SARS-CoV-2.

## 5. Conclusions

In conclusion, in this study we first compared some clinical and biochemical parameters at hospital admission in non-survivor and survivor patients with COVID-19 pneumonia and acute respiratory failure. As expected, among the clinical parameters, respiratory failure was a distinguish signature between the two groups while the most significant difference among biochemical parameters was LDH, as an indicator of tissue and cell damage. Interestingly, the increase in LDH was inversely correlated with P/F as an indicator of respiratory failure. There were also significant differences in other biochemical parameters and cytokines between the two groups. Giving that in our study P/F and some biochemical parameters were found to be correlated with LDH, which in turn explains about 73% of respiratory failure, we hypothesized that at least some of these variables participate in the mechanisms leading to regulated cell death of lung cells and thus to patient’s death. Thus, to identify the most relevant clinical and biochemical features in predicting mortality and LDH levels in COVID-19 patients, we applied a combination of feature selection and predictive modeling techniques, integrating LASSO feature selection with regression models. For mortality prediction, both Cox proportional hazards regression and logistic regression were tested. For LDH prediction, LASSO was combined with linear regression, with performance assessed using Normalized Root Mean Squared Error (NRMSE). To facilitate the application of artificial intelligence techniques utilizing all clinical features and subjects, applying imputation on missing data was necessary. The Miss Forest algorithm was employed for this purpose. By using both LASSO with Cox regression and LASSO with logistic regression for mortality prediction, LDH emerges as the most discriminating feature compared to other variables. Furthermore, when LASSO with Cox regression model was applied, IL-22 emerged as the second most important predictor of mortality while in the LASSO with logistic regression model, IL-10 turned up as the second most significant predictor of mortality. This discordance of results may be dependent the fact that IL-22 and IL-10 belong to the same “IL-10 family” and share the same receptors and activities [56]. In both LASSO models, eGFR and creatinine emerged as further subsidiary predictors of mortality. Finally, by applying the LASSO–linear regression analysis, the most influential predictor was P/F followed by IL-10. Other notable predictors included eGFR, RDW, IL-1Ra, number of monocytes, hepcidin, and age, each contributing moderately to the predictive model. Using this multivariate approach, we identified key features that enhance our ability to differentiate between survivors and non-survivors while also providing insights into the determinants of LDH levels, an important marker in COVID-19 patients. This approach seems to provide a refined understanding of factors influencing mortality and LDH levels, offering valuable insights into disease mechanisms and patients prognosis.

## Figures and Tables

**Figure 1 viruses-17-00359-f001:**
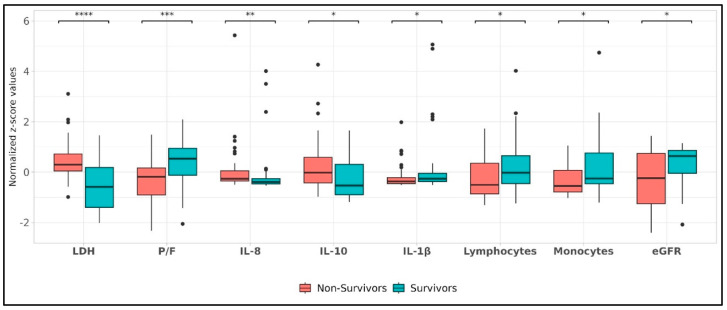
**Significant Wilcoxon parameters**. Boxplots illustrating the distribution of parameters deemed significant by the Wilcoxon test, used to differentiate between the two classes. The *s indicate the level of significance. If 0.01 < *p*-value ≤ 0.05, we indicate it with *, if 0.001 < *p*-value ≤ 0.01 we indicate it with **, if 0.0001 < *p*-value ≤ 0.001 we indicate it with ***, whereas if *p*-value ≤ 0.0001 we indicate it with ****.

**Figure 2 viruses-17-00359-f002:**
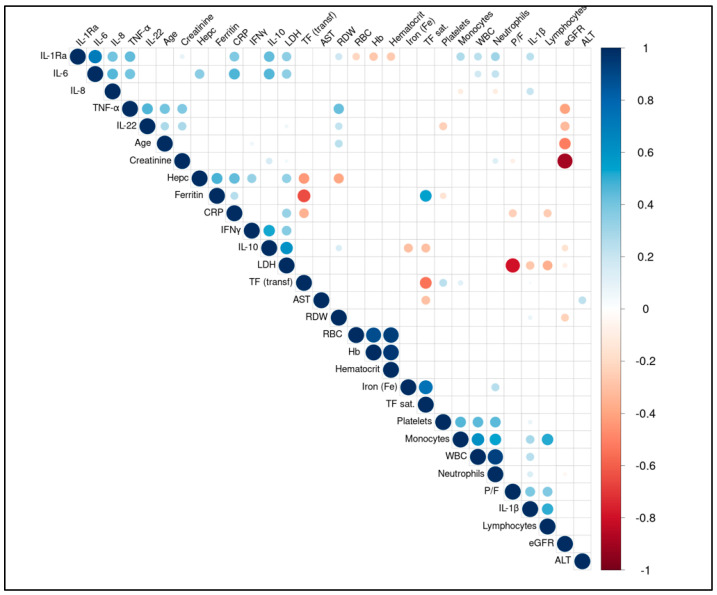
**Significant Spearman correlations among clinical and biochemical variables**. Matrix of Spearman correlations among variables, displaying only significant correlations (*p*-value ≤ 0.05). The size of each circle represents the absolute value of the corresponding correlation coefficient. Variables are ordered according to hierarchical clustering.

**Figure 3 viruses-17-00359-f003:**
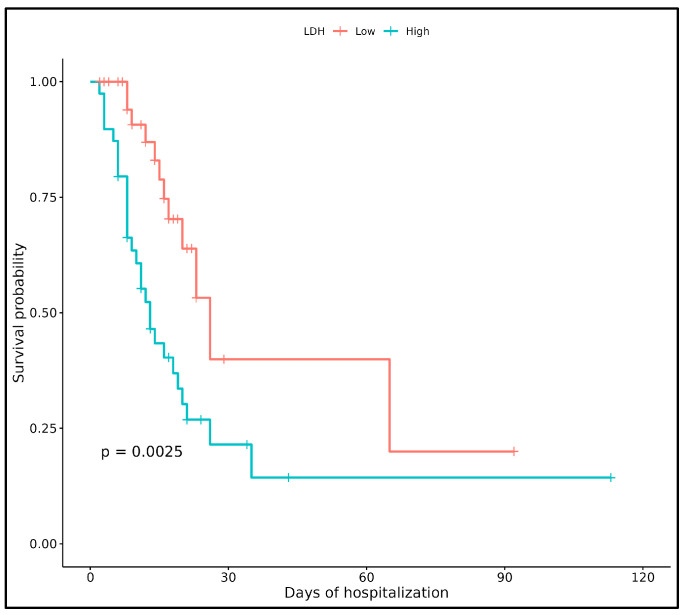
**Kaplan–Meier survival curves for LDH.** Kaplan–Meier survival curves for patients stratified into low (red line) and high (blue line) groups based on the median value of LDH at hospital admission. The y-axis represents survival probability, while the x-axis indicates days of hospitalization. The *p*-value (0.0025) was obtained using the log-rank test, indicating a statistically significant difference in survival between the two groups.

**Figure 4 viruses-17-00359-f004:**
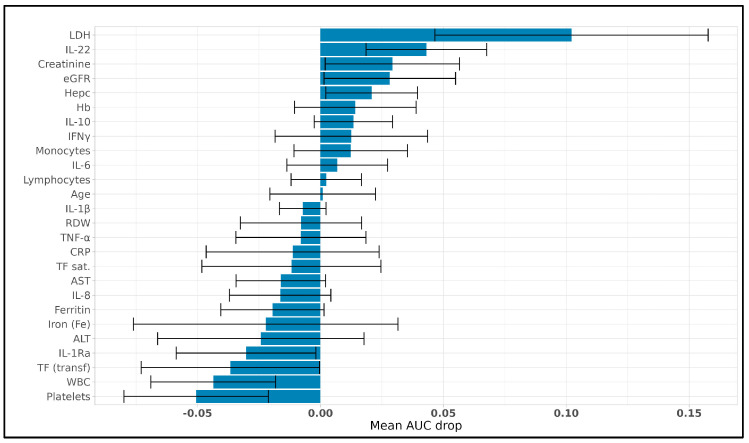
**Permutation importance for LASSO–Cox model predicting mortality.** Permutation importance analysis results showing the mean AUC drop for each feature in predicting mortality using the LASSO–Cox model. Variables with a positive AUC drop (rightward bars) indicate features that, when permuted, led to a decrease in model accuracy, underscoring their predictive importance. LDH demonstrated the highest predictive contribution, followed by IL-22 and creatinine. Variables with minimal or negative AUC drops (leftward bars) showed limited influence, indicating a weaker or negligible independent association with mortality.

**Figure 5 viruses-17-00359-f005:**
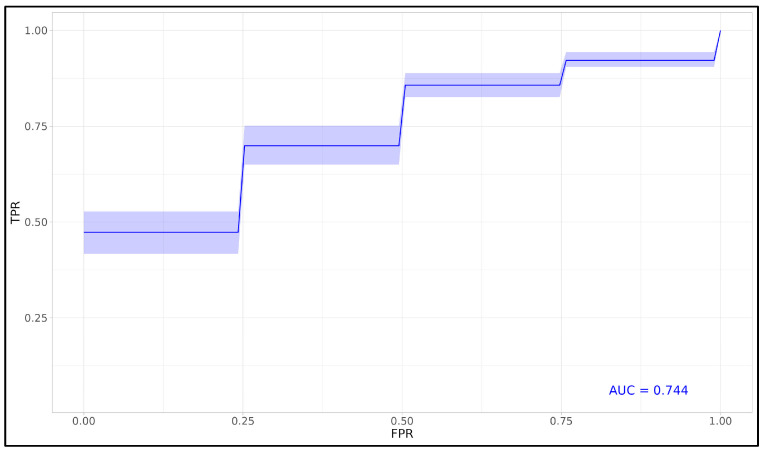
**Mean ROC curve for LASSO–Cox model.** Mean ROC curve derived from 1000 iterations for the final Cox regression model, using the most impactful features identified through the LASSO Deviance configuration on the Reduced Dataset. The plot shows sensitivity versus 1—specificity across various thresholds, with an average AUC of 0.744 and 95% confidence intervals, indicating the model’s predictive accuracy for mortality outcomes. This averaged ROC curve represents the model’s performance across multiple training and testing cycles, providing a robust assessment of its ability to generalize to unseen data.

**Figure 6 viruses-17-00359-f006:**
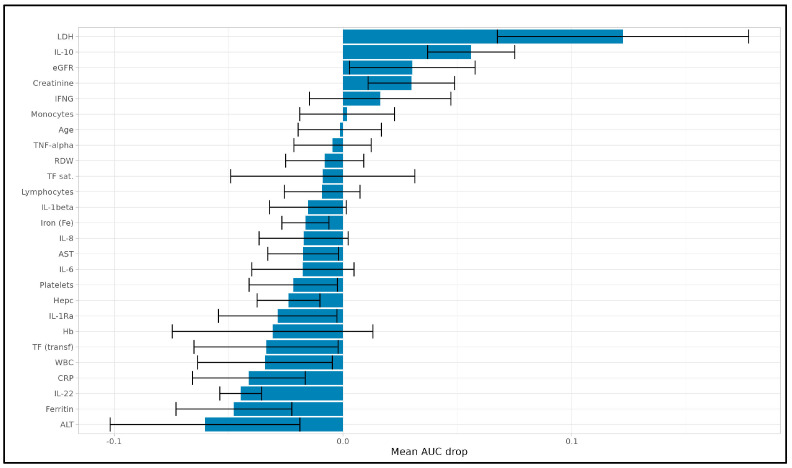
**Permutation importance for LASSO–logistic model predicting mortality.** Permutation importance analysis results showing the mean AUC drop for each feature in predicting mortality using the LASSO–logistic model. Variables with a positive AUC drop (rightward bars) indicate features that, when permuted, led to a decrease in model accuracy, underscoring their predictive importance. LDH demonstrated the highest predictive contribution, followed by IL-10, eGFR, and creatinine. Variables with minimal or negative AUC drops (leftward bars) showed limited influence, indicating a weaker or negligible independent association with mortality.

**Figure 7 viruses-17-00359-f007:**
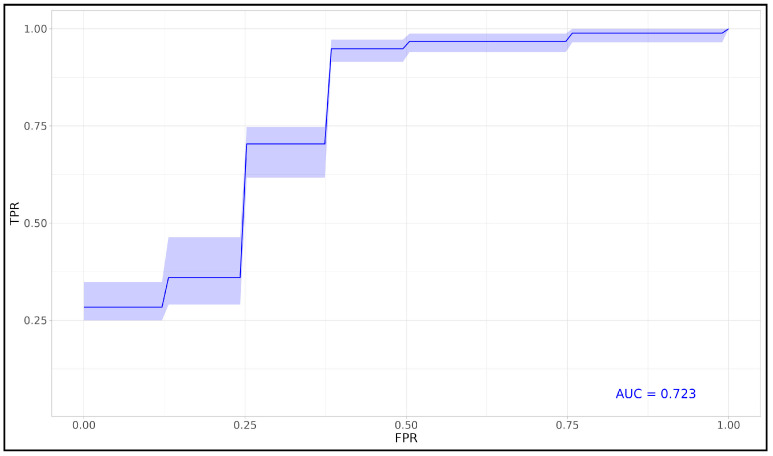
Mean ROC curve for LASSO–logistic model. Mean ROC curve derived from 1000 iterations for the final logistic regression model, using the most impactful features identified through the LASSO Deviance configuration on the Reduced Dataset. The plot shows sensitivity versus 1 - specificity across various thresholds, with an average AUC of 0.723 and 95% confidence intervals, indicating the model’s predictive accuracy for mortality outcomes. This averaged ROC curve represents the model’s performance across multiple training and testing cycles, providing a robust assessment of its ability to generalize to unseen data.

**Figure 8 viruses-17-00359-f008:**
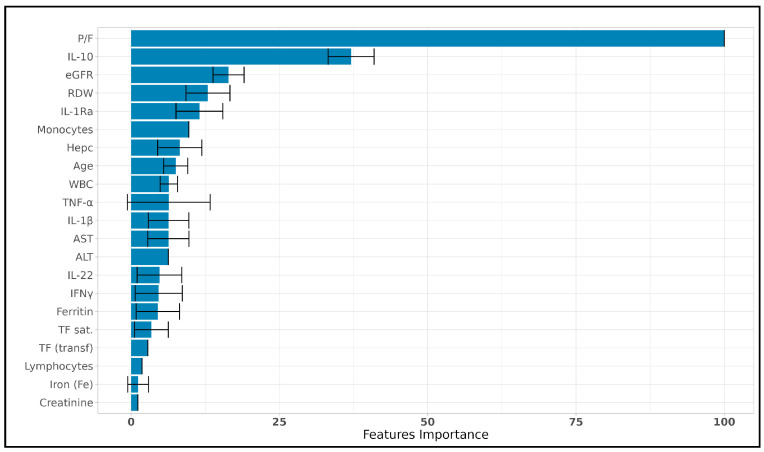
**Feature importance for predicting LDH levels using LASSO and linear regression.** Bar plot showing the relative importance of selected features for predicting LDH levels, as identified by LASSO combined with linear regression. The P/F ratio (PaO_2_/FiO_2_) emerged as the most influential predictor, followed by IL-10, eGFR, and RDW. Error bars represent the standard deviation of the importance scores across 1000 iterations. Features are ordered by decreasing importance, with variables at the bottom showing minimal contribution to the predictive model.

**Table 1 viruses-17-00359-t001:** Demographic, clinical, and biochemical characteristics of the two groups of COVID-19 patients.

	SURVIVORS (n. 40)	NON-SURVIVORS (n. 40)	*p*- Value
**Age**	77.5 [73–84]	80.5 [76.7–86]	NS
**Sex (M/F)**	21/19	21/19	NS
**Onset of symptoms (days)**	5.5 [2–7.7]	8 [5–12]	<0.05
**PaO_2_/FiO_2_ (ratio)**	281 [229–314]	224 [166.5–252]	<0.0005
**Admission to ICU n. (%)**	5 (12.5)	10 (20)	<0.005
**Orotracheal intubation n. (%)**	4 (10)	8 (20)	NS
**Comorbidities**			
Hypertension n. (%)	24 (60)	18 (45)	NS
Diabetes mellitus n. (%)	9 (22.5)	4 (10)	NS
Chronic kidney disease n. (%)	3 (7.5)	5 (12.5)	NS
Chronic liver disease n. (%)	2 (5)	2 (5)	NS
**Biochemical parameters**			
Creatinine (mg/dL)	0.8 [0.7–1]	1.0 [0.7–1.5]	NS
eGFR (mL/min/1.73 m^2^)	81.5 [64.8–87]	60 [35.2–84]	<0.05
AST (U/L)	35 [26–52]	38.5 [30–47]	NS
ALT (U/L)	25 [17.5–47.5]	27 [21.0–37.2]	NS
LDH (U/L)	239.5 [159.8–315]	325.5 [301–367]	<0.00005
CRP (mg/L)	68 [36–126]	78 [46–147]	NS
Iron (mcg/dL)	33.9 [24.6–53.6]	35.1 [26.5–47.8]	NS
Transferrin (g/L)	1.5 [1.3–1.9]	1.6 [1.1–1.9]	NS
Transferrin sat. (%)	16 [11.5–25.5]	16 [11–26.5]	NS
Ferritin (mcg/L)	870 [495–1194]	813.5 [504–1555.2]	NS
Hematocrit (%)	40.5 [34–43]	39.0 [36–43]	NS
Hb (g/dL)	13.1 [11.2–13.7]	12.5 [12–14]	NS
Hepcidine (ng/mL)	199.9 [114.7–244.6]	188.2 [87.7–270]	NS
RBC (10^12^/L)	4.3 [3.8–4.7]	4.2 [3.8–4.7]	NS
RDW (%)	13.5 [12.9–14.8]	14.3 [13–15.8]	NS
Platelets (10^9^/L)	212 [177.8–267.8]	188 [137.5–232]	NS
WBC (10^9^/L)	6.8 [5.2–9.9]	6.5 [4.3–10.6]	NS
Neutrophils (10^9^/L)	5.2 [4–7.1]	5.3 [3.1–9]	NS
Lymphocytes (10^9^/L)	0.7 [0.5–1]	0.5 [0.4–0.9]	<0.05
Monocytes (10^9^/L)	0.34 [0.3–0.7]	0.2 [0.2–0.4]	<0.05

Data are presented as n (%) or median [interquartile range]. The Wilcoxon test was employed to assess the statistical significance of differences between the groups of survivors and non-survivors, with a significance level of 5% (*p* ≤ 0.05). For comorbidities and intubation status, the chi-squared test was utilized. ALT: alanine aminotransferase; AST: aspartate aminotransferase; CRP: C-reactive protein; eGFR: estimated glomerular filtration rate (CKD formula); ICU: Intensive Care Unit; PaO_2_: arterial partial oxygen tension; FiO_2_: inspiratory oxygen fraction; NS: not significant.

**Table 2 viruses-17-00359-t002:** Plasma concentrations and comparison of cytokines in the two groups of COVID-19 patients.

Biomarker (pg/mL)	SURVIVORS (n. 40)	NON-SURVIVORS (n. 40)	*p*-Value
IFNγ	1.9 [0.5–7.5]	1.6 [0.8–6.4]	NS
IL-1β	0.3 [0.2–0.5]	0.2 [0.1–0.3]	<0.05
IL-1Ra	995 [700–1792]	1179 [765–2271]	NS
IL-10	9.5 [4.7–20.5]	16.2 [10.9–24.1]	<0.05
IL-22	16.5 [10.4–26.7]	16.1 [10.5–42.6]	NS
IL-8	6.9 [4.8–10.6]	10.5 [8–19.1]	<0.005
IL-6	19.6 [7–44.9]	29.2 [11.9–62.4]	NS
TNF-α	13.6 [11.3–16.1]	14.8 [11.6–21]	NS

Data are presented as median [interquartile range]. The Wilcoxon test was employed to assess the statistical significance of differences between the groups of survivors and non-survivors, with a significance level of 5% (*p* < 0.05). NS: not significant.

**Table 3 viruses-17-00359-t003:** Significant Spearman correlation analysis of LDH with other parameters.

Feature 1	Feature 2	Correlation	*p*-Value
LDH	IL-10	0.6066	3.1 × 10^−9^
LDH	IFNγ	0.3614	9.9 × 10^−4^
LDH	IL-6	0.3458	1.8 × 10^−3^
LDH	IL-1Ra	0.3405	2.0 × 10^−3^
LDH	Hepcidine	0.3333	2.5 × 10^−3^
LDH	CRP	0.3228	5.7 × 10^−3^
LDH	IL-1β	−0.2760	1.3 × 10^−2^
LDH	Iron (Fe)	−0.2882	9.5 × 10^−3^
LDH	Lymphocytes	−0.3633	1.2 × 10^−3^
LDH	P/F	−0.7929	0.0

Significant Spearman correlations between various parameters and LDH levels. Correlation coefficients (r) and corresponding *p*-values are reported. A *p*-value ≤ 0.05 indicates statistical significance.

## Data Availability

The data presented in this study are available on request from the corresponding author.

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
