# Peer review of "Identification of Early Biomarkers of Mortality in COVID-19 Hospitalized Patients: A LASSO-Based Cox and Logistic Approach"

_viruses, 2025, doi:10.3390/v17030359_

Round 1

Reviewer 1 Report (Previous Reviewer 1)

Comments and Suggestions for Authors

We greatly appreciate the authors’ thoughtful responses to our previous comments and their considerable efforts in revising the manuscript. The expanded analyses and clearer explanations of the methods and results are commendable. In light of the additional information provided, we offer the following suggestions:

1.  While the manuscript’s title has been changed to “A machine-learning approach to detect early biomarkers of mortality in COVID-19 hospitalized patients,” the primary focus of the study appears to be the identification of key biomarkers via LASSO feature selection combined with Cox proportional hazards and logistic regression models. Therefore, we recommend emphasizing the biomarker discovery process in the title, rather than centering it solely on the machine-learning approach. Highlighting the process of biomarker mining will more accurately reflect the study’s core contributions.

2.  The legends of Figures 5 and 6 could be further refined.

Author Response

Comments and Suggestions for Authors

We greatly appreciate the authors’ thoughtful responses to our previous comments and their considerable efforts in revising the manuscript. The expanded analyses and clearer explanations of the methods and results are commendable. In light of the additional information provided, we offer the following suggestions:

  1. While the manuscript’s title has been changed to “A machine-learning approach to detect early biomarkers of mortality in COVID-19 hospitalized patients,” the primary focus of the study appears to be the identification of key biomarkers via LASSO feature selection combined with Cox proportional hazards and logistic regression models. Therefore, we recommend emphasizing the biomarker discovery process in the title, rather than centering it solely on the machine-learning approach. Highlighting the process of biomarker mining will more accurately reflect the study’s core contributions.

According to the suggestion, the title of the manuscript has been changed to: Identification of Early Biomarkers of Mortality in COVID-19 Hospitalized Patients: A LASSO-Based Cox and Logistic Approach

  1. The legends of Figures 5 and 6 could be further refined.

We thank the reviewer for the suggestion. We have extended the legends of Figures 5 and 6 (which are now Figures 6 and 7 in the revised version, following the addition of a new figure to address another reviewer’s comment). Additionally, thanks to your observation, we identified and corrected an error in the title of the previous Figures 3 and 6 (now Figures 4 and 7). The revised legends and figure titles are now accurate and more detailed

Reviewer 2 Report (Previous Reviewer 2)

Comments and Suggestions for Authors

I appreciate the work conducted to identify early biomarkers of mortality in hospitalized COVID-19 patients using advanced statistical models. However, there are critical methodological concerns regarding how survival has been treated in the study.

1.Survival Treated as a Non-Time-Dependent Variable

In this study, survival is treated as a binary outcome (survivor/non-survivor) using logistic regression. While useful for classification, this approach completely disregards the time-to-event component, which is crucial in survival studies for accurately assessing patient prognosis.

Although the Cox proportional hazards model was employed, there is no clear discussion on whether the proportional hazards assumption was validated, which is fundamental for the reliability of the Cox model results.

Please treat a time dependent outcome not as binary variable.

2.Implications of This Simplification

Ignoring the temporal dimension may introduce interpretative bias and limits the model’s ability to capture the actual dynamics of mortality over time.

Treating survival as static risks underestimating the effect of certain predictors, which may exhibit significant impact over the long term.

3.Suggestions to Improve the Study

While time-dependent covariates were not considered, I recommend at least including:

Kaplan-Meier survival curves to compare the distribution of survival times between patient groups.

A more detailed discussion on the implementation and validation of the Cox model, particularly addressing the proportional hazards assumption.

Author Response

We have considered your suggestions (see the attached file) and modified the manuscript accordingly (text in red).

This manuscript is a resubmission of an earlier submission. The following is a list of the peer review reports and author responses from that submission.

Round 1

Reviewer 1 Report

Comments and Suggestions for Authors

The study aims to identify early biomarkers that could predict mortality among hospitalized COVID-19 patients using Recursive Feature Elimination via Random Forest (RFE-RF). The authors highlight the significance of LDH, eGFR, IL-8, creatinine, IL-10, hepcidin, the number of monocytes, the number of lymphocytes, and age as biomarkers for early prediction of mortality in COVID-19 patients. This has the potential to lead to improved management strategies and outcomes for these patients.

However, the following issues need to be addressed:

1. Justification for Method Selection: The article employs RFE-RF as the machine learning method for detecting early biomarkers of mortality in hospitalized COVID-19 patients. The rationale behind choosing this specific method needs further elaboration. Is this method demonstrably superior to other machine learning methods of a similar type?

2. Final Model Construction: In the construction of the final model, the authors include 9 biomarkers in addition to P/F and Hb, derived from the most accurate survival prediction classification model based on RFE-RF. However, Hb does not show strong correlation with other features and does not exhibit significant differences (as indicated in Table 1 and Figure 2). The inclusion of P/F and Hb in the model warrants a clearer justification to explain how these additions contribute to the model's performance.

3. External Validation: The credibility of the model could be significantly enhanced by validating the findings using an external dataset. This step is crucial for verifying the robustness and generalizability of the model.

Reviewer 2 Report

Comments and Suggestions for Authors

I do not understand why a transformation of the data should be implemented, since machine learning is a data-driven method that does not assume basic assumptions. The problem might only be related to sparsity and variability.

Random forest is not an adequate model for survival; use survival random forest. The sample is too small for the algorithm to learn. The accuracy produced is only the result of overfitting. This methodology is not appropriate for the sample presented.

After expanding the sample use other classifiers and produce relative AUCs.

After expanding the sample use other classifiers and produce relative AUCs to compare the methods.

I am pretty sure that a logistic regression is a good classifier in this case, although when we talk about survival, we are talking about time-dependent events, and so we have to use the COX model.

Reviewer 3 Report

Comments and Suggestions for Authors

Article easy to read, well described, well supported by a large and relevant choice of references. A good description of methods has been done. The analysis of all data and parameters is methodical, so that results corroborate numbers of previous studies. This has been well analysed in discussion.

Nethertheless,  the article presents some weaknesses. First of all, this is a retrospective study. Is it worth in 2024 to produce an retrospective article about COVID-19 data collected in 2020-2022? The methodology is harder to establish and there is more probability of missing data. It can be the reason for using Miss Forest Algorithm (by RFE-RF method). But Miss Forest Algorithm is not always the best for analysis, sometimes it can provide some biais (cf articles as: DOI: 10.1186/s12874-020-01080-1; DOI: 10.1186/s12874-021-01272-3 or DOI: 10-1016/j-ya-2020.11-010). In this paper, are we sure that number of subjects sufficient to use that algorithm? Then, the missing data (table S1) are important only for transaminases (almost 50%), CRP and monocytes. Except monocytes, the other parameters are not crucial for discussion. Would conclusions have been really changed without the algorithm?

Considering the Introduction: what is the aim of this study? No question has been clearly asked.

Caution in the Results section:

There are some errors to be reconciliate between text and Table 1:

-            lines 179-180: Median time from onset of COVID symptoms: there is an inversion in Table 1 columns (noted 5.5 days in survivors and reverse in the text) Values in the text seem to be correct

-            lines 181-182: P/F values not so precise in the text, have to be more accurate: 281 for survivors and 224 for non survivors (correct in Table 1)

-            line 186: Admission in ICU, p < 0.005 difference is significant (from Table 1)

Table 2 and lines 213-215:

-            for better fluent reading, data in text and Table 2 should be enounce in the same order

Conclusion is drastically short! This directly comes from the introduction with no question asked. Again what is really the aim of this study? The article shows congruent results in respect to literature on the subject, but this is not original at all, regarding the many articles produced in 4 years. Should the aim could be demonstrating the RFE-RF validity method, as the title seems to mention? The discussion and conclusion should be refounded through this question.

So, the article can not be published at that time.